# Effects of Vernalization on Off–Season Flowering and Gene Expression in Sub-Tropical Strawberry cv. Pharachatan 80

**Thanyarat Thammasophon, Tonapha Pusadee** 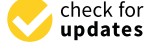**, Weenun Bundithya and Daruni Naphrom ***

Department of Plant and Soil Sciences, Faculty of Agriculture, Chiang Mai University,
Chiang Mai 50200, Thailand
* Correspondence: daruni.n@cmu.ac.th; Tel.: +66-085-0407747

**Abstract:** Off-season strawberry production may diversify the yield, thereby increasing costs, but the environmental conditions are a limiting factor. This experiment aimed to study the effects of vernalization on off-season flowering and gene expression in sub-tropical strawberry cv. Pharachatan 80. The factorial $(2 \times 2) + 1$ in a completely randomized design was used in this study. Factor A was the vernalization temperatures: 2 °C and 4 °C. Factor B was the vernalization periods: 1 week and 2 weeks, compared with non-vernalization (control). The expression profile of genes was determined after vernalization treatments. The results revealed an interaction between the two factors on the number of days it took the plants to bloom, the percentage of flowering, the number of inflorescences, the number of flowers per inflorescence and the number of flowers per plant, whereas the number of first flower bloom days, inflorescence length and flower size were not affected by the interaction between the two factors. Strawberry plants vernalized for 1 and 2 weeks at 2 °C showed earlier flowering (21.4 and 23.1 days, respectively) than did those vernalized at 4 °C (24.9 and 25.7 days, respectively). On the other hand, non-vernalized strawberry plants took longer to bloom, at 62.2 days. Strawberry plants vernalized at 2 °C for 2 weeks had the highest percentage of flowering, number of inflorescences, number of flowers per inflorescence and number of flowers per plant. The analysis on gene expression showed that VRN5, SOC1 and FT genes were upregulated after vernalization at 2 °C for 2 weeks, whereas gene expression of the control treatment was not detected. This study demonstrates that vernalization treatment could induce off-season flowering in sub-tropical strawberry cv. Pharachatan 80 by activating flowering genes.

**Keywords:** floral induction; flowering genes; vernalization



## 1. Introduction

From an economic perspective, strawberries are among the most important fruit produced around the world. The consumption of fresh fruits and industrial processing have increased rapidly. In Thailand, strawberry planting areas are mainly located in the north and the northeast highlands due to the cool weather there. The strawberry cv. Pharachatan 80 is a major fresh consumption variety in Thailand. It is classified as a short-day type and requires a cold temperature, 10–15 °C, for 20–30 days to induce flower budding [1]. Thailand is located in a sub-tropical region consisting of three seasons: a summer season from the middle of February to the middle of May, a rainy season from mid-May to mid-October, and a cold season taking place from mid-October to mid-February and has an average annual temperature of 28 °C [2].

Strawberries can be produced from only one crop with a short harvesting period in the cold season, leading to oversupply and, subsequently, lower pricing [3]. Off-season fruit production is an ideal solution in Thailand, providing the opportunity to manipulate flowering and fruit setting periods. The successful off-season flowering cases are mostly evident in fruit trees such as mango, longan and durian, in which plant growth retardants are used to overcome environmental conditions; however, the same strategy is not used

in small-fruit crops such as strawberries. Vernalization, using low temperatures to induce floral buds in plants, was reported to accelerate flowering in these kinds of small-fruit crops. A low temperature treatment at 10 °C for 100–400 h stimulated flower buds and increased the flowering volume of the strawberry plant [4]. Acclimatization of strawberry plants with low temperature is a simple method used to manipulate and promote off-season flowering.

Floral mechanisms are controlled by several genes that respond to the prevailing environmental conditions. In the *Arabidopsis thaliana* plant, expression of miR172 in leaves activates FLOWERING LOCUS T (FT) [5,6], the final output of the photoperiodic pathway [7], through repression of AP2-like transcripts SCHLAFMÜTZE (SMZ), SCHNARCHZAPFEN (SNZ) and TARGET OF EAT 1–3 (TOE1–3) [6,8]. Meanwhile, an increase in the expression level of the SPLs gene at the shoot apical meristem (SAM), leads to the transcription of floral meristem identity (FMI) genes [9,10]. The FMI genes trigger the expression of floral organ identity genes [11], which function in a combinatorial fashion to specify the distinct floral organ identities. The FT and SOC1 genes act as floral integrators, which receive trigger signals and thus initiate flower induction [12]. MADS-box gene FLC (Flowering Locus C) prevents flowering by repressing FT and SOC1, and vernalization is needed to nullify its function [13]. During vernalization, FLC is downregulated by VRN2-PRC2 (Vernalization 2-Polycomb Repressive Complex 2) protein complex containing low-temperature-activated VIN3 (VERNALIZATION INSENSITIVE3), allowing plants to flower [14,15].

Gene expression analysis in strawberry plants, which was carried out by Mouhu et al. (2009) [16], indicated that two putative flowering genes, AP1 and LFY, were co-regulated during floral development in ever-bearing (EB) wild strawberry. The homolog of floral identity gene AP1 was expressed in the EB wild strawberry apex at a particular leaf stage and its expression was strongly enhanced during later developmental stages. LFY mRNA also accumulated along with AP1 during floral development in the EB genotype, whereas SOC1 did not show a clear trend. The mRNAs of SOC1 and LFY were also present in the short-day (SD) genotype but no AP1 transcription was detected. Badex et al. (2015) [17] investigated the gene expression of cold-susceptible strawberry cultivar 'Elsanta' and the cold-tolerant cultivar 'Selvik' at 0, 6 and 12 weeks after sub-zero treatment at −12 °C and found that the F3H transcript in the treated 'Selvik' plants reached a level four times higher than that of control plants at the 12th week post-treatment. In 'Elsanta', the CBF4 and COR47 genes were slightly upregulated in the sixth week after treatment, whereas the F3H gene remained stably expressed. High positive correlation between the transcript level of the COR47 and CBF4 genes was observed in both cultivars.

The varying gene expressions in strawberry depend on varieties and environmental conditions, especially in relation to the duration and degree of the low temperatures. It was reported that vernalization induced VRN5, which repressed FLC expression, causing flower induction due to FT and SOC1 activation [16]. Therefore, in this research, these genes were investigated to determine the effect of low temperature and optimal period for inducing flowering via the vernalization pathway.

In Thailand, there have been several studies on disease-free propagation, yield and fruit quality improvement, and post-harvest management of strawberries but few studies about gene expression related to flower induction. Therefore, this study investigated the use of low temperature to induce off-season flowering and gene-expression responses. The results of this experiment can be used to improve off-season strawberry production and provide gene expression data for sub-tropical strawberry molecular breeding in the future.

## 2. Materials and Methods

### 2.1. Plant Materials

The cultivar known as strawberry *Fragaria × ananassa* cv. Phrachatan 80 was selected from hybridization of Japanese strawberry cv. Royal Queen under the strawberry improvement program of the Royal Project Foundation during 1998–2002 at 18°48′39″ N, 98°53′5″ E, Chiang Mai province, Thailand.

## 2.2. Growing Conditions

The experiment was conducted in greenhouses at the Faculty of Agriculture, Chiang Mai University (18.7934 N, 98.9600 E), Chiang Mai province, Thailand during February to May 2021. Uniform strawberry plants of the cv. Pharachatan 80 (450 plants in total) were grown in 3-inch pots containing composted coconut husk. After that, they were placed into a growth chamber and then vernalized at 2 °C and 4 °C for 1 and 2 weeks, with an 8 h light (102.5 µmol/m2s1 photosynthetic photon flux density; PPFD)/16 h dark photoperiodic cycle. Meanwhile, the non-vernalized strawberry plants were kept and grown under ambient greenhouse conditions. After 2 and 4 weeks of growth, the vernalized strawberry plants were placed in ambient greenhouse conditions. They were transferred to 8-inch pots containing composted coconut husk mixed with peat moss at a ratio of 2:1 by volume. The leaves of non-vernalized plants were trimmed off to match the number of leaves on the vernalized plants, then plant growth and flowering were investigated.

## 2.3. Gene Expression Analysis

### 2.3.1. RNA Extraction

Total RNA was extracted from strawberry crowns following the manufacturer's guidelines for Trizol® (Invitrogen, Waltham, MA, USA). Extracted RNA samples were quantified using a nanodrop spectrophotometer.

### 2.3.2. Treating DNase I

The total RNA was treated with DNase I, RNase-free (Thermo scientific, Waltham, MA, USA) under the following conditions: 1 µg of total RNA, 2 µL of 10× reaction buffer, 1 µL of DNase I, Rnase-free and 17 µL of DEPC-treated water (total volume 20 µL). The reaction was incubated at 37 °C for 30 min. Next, we added 1 µL of 50 mM EDTA and incubated the mixture at 65 °C for 10 min.

### 2.3.3. cDNA Synthesis

PCR was carried out using 1 µg of total RNA (Dnase I-treated) for cDNA synthesis using a Tetro cDNA Synthesis Kit (Thermo scientific, Waltham, MA, USA) with the following conditions: 1 µL of oligo (dT)18, 1 µL of 10 mM dNTP mix, 4 µL of 5× RT buffer, 1 µL of RiboSafe Rnase Inhibitor, 1 µL of Tetro Reverse Transcriptase (20 U/µL) and DEPC-treated water; the total volume was 20 µL. The reaction was incubated at 45 °C for 30 min and then terminated by incubating at 85 °C for 5 min. The mixture was then stored at −20 °C. The cDNAs were verified for quantity and quality using a nanodrop spectrophotometer and 1.5% agarose gel electrophoresis.

### 2.3.4. Gene Expression by Semi-Quantitative RT-PCR Analysis

Gene expression levels of VRN5, FT and SOC1 were analyzed via semi-quantitative RT-PCR using gene-specific primers for VRN5, FT, SOC1 and UBIsw (housekeeping gene). PCR was performed in triplicate for amplification of cDNA templates with VRN5, FT, SOC1 and UBIsw with the following reaction: 1 µL cDNA was placed in PCR tubes of 0.2 mL volume, with 8 µL of water (ddH2O), 1 µL of 10× Reaction Buffer, 0.5 µL 50 mM MgCl$_2$, 0.1 µL of forward and reverse primers and 0.1 µL of 5-unit MyTaqTM HS DNA Polymerase (Bioline, London, UK). Then, 0.1 µL of 2.5 mM dNTP and 0.3 µL of DMSO were added.

The PCR was performed by denaturing at 95 °C for 2 min followed by 40 cycles at 95 °C for 30 s, primer annealing for 30 s (depending on the suitability of each primer) (Table 1), extension at 72 °C for 30 sec and a final extension at 72 °C for 5 min.

**Table 1.** Primer used for expression study of SOC1, VRN5, FT and UBISw genes.

| Primer | Annealing Temperature (°C) | DNA Sequence (5′ to 3′) | Reference |
|---|---|---|---|
| SOC1 | 49 | F: ACTTGCTGGGTTCATTTTCC | [18] |
| | | R: GAGCTTTCCTCTGGGAGAGA | |
| VRN5 | 48 | F: AGCCCTTGATGTCATCAGCTG | [16] |
| | | R: CCGATGAATGGTTGGCTAATG | |
| FT | 53 | F: CAATCTCTTGGCCGAAAACT | [18] |
| | | R: TGAGGCTCAAACCTTCCCAAG | |
| UBISw | 53 | F: CAGACCAGCAGAGGCTTATCTT | [16] |
| | | R: TCTGGATATTGTAGTCTGCTAGGG | |

*2.4. Data Analysis*

　　Statistically comparative data were analyzed by analysis of variance, and mean difference was compared using LSD at a 95% confidence level.

　　The related transcript quantification was performed using relative intensity to reference gene (UBIsw). Gene expression levels were analyzed by relative intensity to the reference gene using ImageJ software version 1.50i (Wayne Rasband National Institutes of Health, Bethesda, MD, USA). The relative intensity of gene expression was subjected to statistical analysis using Statistica 9 software (analytical software SX, version 9, Tallahassee, FL, USA).

$$\text{Relative intensity} = \frac{\text{Intensity of DNA}}{\text{Intensity of Actin}}$$

## 3. Results

*3.1. Effect of Vernalization and Flowering*

　　This experiment shows that there was an interaction between the two factors in terms of number of days to flowering and percentage of flowering. Strawberry plants exposed to 2 °C for 1 and 2 weeks showed earlier flowering than those exposed to 4 °C for 1 and 2 weeks. However, all vernalization treatments promoted earlier flowering than the non-vernalized treatments. Vernalization at 2 °C for 2 weeks resulted in the highest percentage of flowering, 92%. The other vernalization treatments showed no difference in the percentage of flowering (60–80%), whereas non-vernalization induced only 10% flowering. There was no interaction between the two factors in terms of the number of days to the first bloom. All treatments were non-significantly different, with a duration of 16.9–18.5 days to the first bloom. However, following vernalized treatments the flower tends to bloom faster than in non-vernalized plants (Table 2) (Figure 1). Table 3 shows an interaction between the two factors in terms of the numbers of inflorescences per plant, flowers per inflorescence and flowers per plant. The strawberry plants exposed to 2 °C for 2 weeks had the highest number of inflorescences per plant, flowers per inflorescence and flowers per plant. Interestingly, all vernalized treatments resulted in significantly more flowering than the non-vernalized treatments. Inflorescence length and flower size were unaffected by the interaction between the two factors. Vernalized strawberry plants had greater inflorescence length and flower size than non-vernalized plants (Table 4).

**Table 2.** Effects of vernalization on number of days to flowering, percentage of flowering and number of days to the first bloom in strawberry cv. Pharachatan 80.

| Treatments | Number of Days to Flowering | Percentage of Flowering | Number of Days to the First Bloom |
|---|---|---|---|
| Vernalization at 4 °C for 1 week | 25.7 [b] | 60.0 [c] | 16.9 |
| Vernalization at 4 °C for 2 weeks | 24.9 [b] | 80.0 [b] | 17.4 |
| Vernalization at 2 °C for 1 week | 21.4 [c] | 80.0 [b] | 17.4 |
| Vernalization at 2 °C for 2 weeks | 23.1 [bc] | 92.0 [a] | 17.2 |
| Non-vernalization | 62.2 [a] | 10.0 [d] | 18.5 |
| Factor 1 | * | * | ns |
| Factor 2 | ns | * | ns |
| Factor 1 × 2 | * | * | ns |

Note: Means followed by different lowercase letters within the same column are significantly different at $p < 0.05$ (ns = non-significant; * = significant).

**Table 3.** Effects of vernalization on number of inflorescences, number of flowers per inflorescence and number of flowers per plant in strawberry cv. Pharachatan 80.

| Treatment | Number of Inflorescences per Plant | Number of Flowers per Inflorescence | Number of Flowers per Plant |
|---|---|---|---|
| Vernalization at 4 °C for 1 week | 2.6 [b] | 2.2 [b] | 6.1 [c] |
| Vernalization at 4 °C for 2 weeks | 2.9 [b] | 2.6 [b] | 7.9 [b] |
| Vernalization at 2 °C for 1 week | 2.7 [b] | 2.2 [b] | 6.2 [bc] |
| Vernalization at 2 °C for 2 weeks | 3.6 [a] | 3.4 [a] | 12.3 [a] |
| Non-vernalization | 1.0 [c] | 1.2 [c] | 1.2 [d] |
| Factor 1 | * | * | * |
| Factor 2 | * | * | * |
| Factor 1 × 2 | * | * | * |

Note: Means followed by different lowercase letters within the same column are significantly different at $p < 0.05$ (* = significant).

**Table 4.** Effects of vernalization on inflorescence length and flower size in strawberry cv. Pharachatan 80.

| Treatments | Inflorescence Length (cm.) | Flower Size (cm) | |
|---|---|---|---|
| | | Length | Width |
| vernalization at 4 °C for 1 week | 6.9 [a] | 2.1 [a] | 2.1 [a] |
| vernalization at 4 °C for 2 weeks | 8.3 [a] | 2.2 [a] | 2.2 [a] |
| vernalization at 2 °C for 1 week | 7.8 [a] | 2.3 [a] | 2.2 [a] |
| vernalization at 2 °C for 2 weeks | 10.3 [a] | 2.1 [a] | 2.3 [a] |
| Non-vernalization | 3.1 [b] | 1.7 [b] | 1.6 [b] |
| Factor 1 | * | ns | ns |
| Factor 2 | * | ns | ns |
| Factor 1 × 2 | ns | ns | ns |

Note: means followed by different lowercase letters within the same column are significantly different at $p < 0.05$, (ns = non-significant; * = significant).

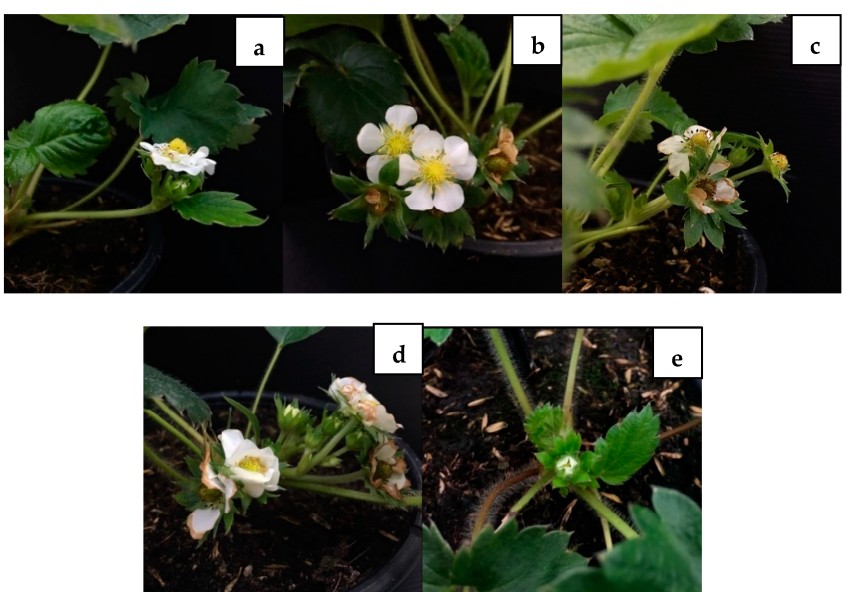

**Figure 1.** Inflorescence of strawberry cv. Pharachatan 80. (**a**) After 4 weeks of vernalization at 4 °C for 1 week. (**b**) After 4 weeks of vernalization at 4 °C for 2 weeks. (**c**) After 4 weeks of vernalization at 2 °C for 1 week. (**d**) After 4 weeks of vernalization at 2 °C for 2 weeks. (**e**) After planting for 8 weeks (control).

### 3.2. Vernalization and Gene Expression

Gene expression by semi-quantitative RT-PCR for three flowering genes, SOC1, FT and VRN5, was investigated. The vernalization revealed cDNA bands when viewed under a UV transilluminator (samples 1–4), whereas the non-vernalization (sample 5) did not (Figure 2).

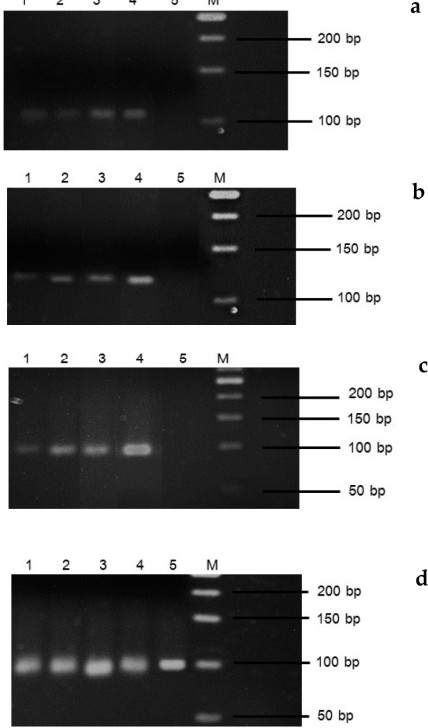

**Figure 2.** Expression of flowering genes in strawberry cv. Pharachatan 80 after vernalization. (**a**) FT, (**b**) SOC1, (**c**) VRN5, (**d**) UBISw (lane 1) 4 °C for 1 week, (lane 2) 2 °C for 1 week, (lane 3) 4 °C for 2 weeks, (lane 4) 2 °C for 2 weeks, (lane 5) no vernalization control, (lane M) DNA ladder.

### 3.3. Expression Levels of VRN5 FT and SOC1 Genes

The results show an interaction between the two factors. Vernalization at 2 °C for 2 weeks showed the highest expression levels of VRN5, FT and SOC1 genes. The vernalization at 4 °C for 2 weeks showed high expression levels for those genes. Notably, between 1 and 2 weeks vernalization, the shorter period induced less VRN5, FT and SOC1 gene expression. On the other hand, the flowering genes could not be detected in non-vernalization treatment (control) (Figure 3).

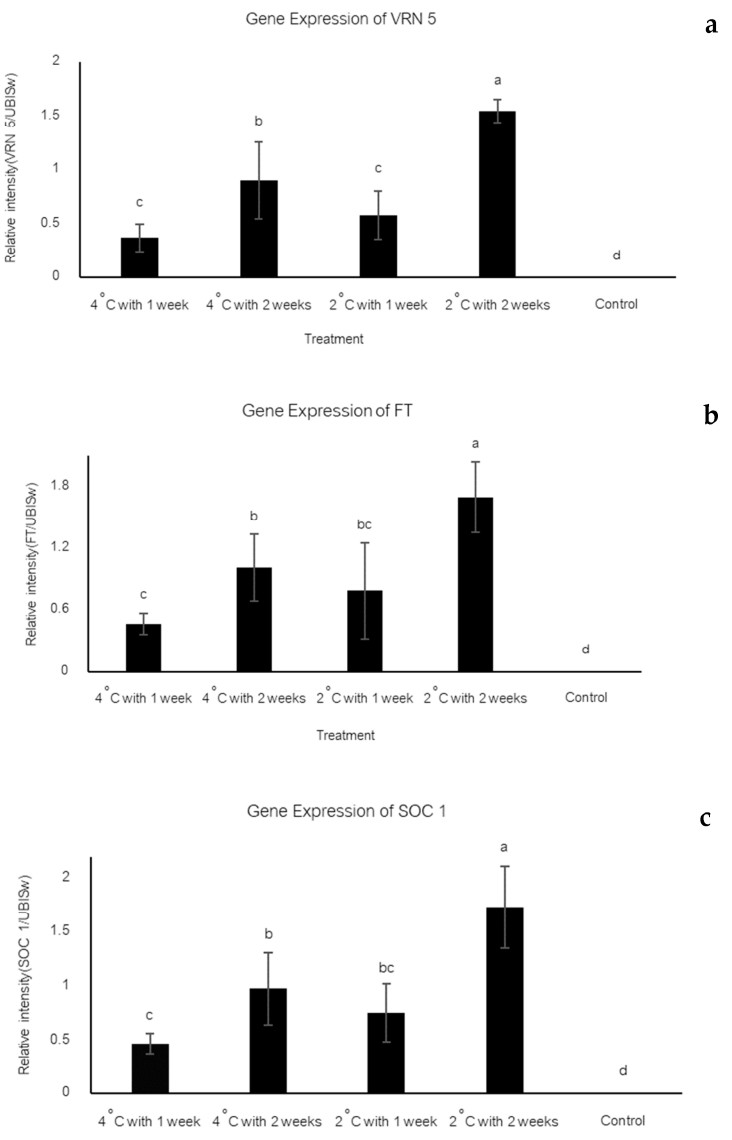

**Figure 3.** Relative expression level of flowering gene expression in strawberry cv. Pharachatan 80 after vernalization. Relative intensity was determined in comparison with that of UBISw reference. (**a**) The relative intensity of VRN5/UBISw, (**b**) the relative intensity of SOC1/UBISw and (**c**) the relative intensity of FT/UBISw.

## 4. Discussion

Sub-tropical strawberries require fewer cold hours than temperate strawberries in order to induce flower buds: 1–3 weeks [19]. Long periods of vernalization result in plant dormancy instead of inducing flower buds. Inappropriate temperatures (too high or too low temperature) lead to inferior development of the apical meristem, resulting in slower or no flowering [20]. In this experiment, all vernalization treatments promoted

off-season flowering in strawberry cv. Pharachatan 80 plants. A lower temperature with longer period, 2 °C for 2 weeks, promoted more flowering than a shorter, 1-week period of vernalization. It seemed that the higher temperature of 4 °C necessitated a longer period of vernalization to promote flowering, whereas non-vernalization did not. Previous studies also indicated that vernalization of Yemen native strawberry varieties at 2 °C for 15 days during the first growing season resulted in a greater number of flowers than non-vernalized treatments. The chilling hour requirement is the main factor that induces strawberry flowering: too little or too much can damage the plants and decrease flowering and fruiting [21]. In addition, vernalization of red raspberry 'Polka' at 6 °C resulted in a higher percentage of flowering than at 18 °C. Low temperatures led to faster flowering than high temperatures [22]. Therefore, a low temperature plays a crucial role in activating the flowering genes and increasing flowering capacity [23].

Our study showed no variation in the number of days to the first bloom after exposure to low temperatures between 16.9 and 18.5 days. In previous experiments, studies on the cofactors between low temperatures and cytokinins in strawberry cv. Pharachatan 72 showed no interaction between low temperature and cytokinins in relation to the number of days to the first bloom [24]. Grez et al. (2020) [25] proposed chilling periods for flowering induction of Chilean white strawberry in a cold room at 2 °C for 10, 20, 30 and 40 days. The flower size and the length of inflorescences in each treatment were not statistically different. It was concluded that the chilling periods had no effect on the days to the first bloom, flower size and the length of inflorescence.

In this experiment, the increase in expression levels of three flowering genes, VRN5, FT and SOC1, in strawberry cv. Phrarachatan 80 was revealed following vernalization treatments. The levels were highest when plants were treated at 2 °C for 2 weeks and the second highest was at a temperature of 4 °C for 2 weeks. The experiment also demonstrated the involvement of the three genes in the flowering process. VRN5 functions in vivo to regulate FLC, the histone-modification profiles of VRN5 mutants were analyzed by chromatin immunoprecipitation. VRN5 mutants' reduction in vernalization induced histone H3 deacetylation, supporting the function in vivo. The interaction and histone-modification data suggest a model for vernalization: the prolonged cold induced at the shoot apex heterodimerizes with VRN5. Unusual expression of other targets of VRN5 may contribute to flowering time [26]. The vernalization pathway has a MADS-box transcription factor FLOWERING LOCUS C (FLC) that inhibits flowering, and it is controlled by FRIGIDA (FRI) in non-vernalized plants. FLC delays flowering by binding to the regulatory regions of several genes encoding floral activators: the FT gene in the leaves and the SOC1 gene. After vernalization, FLC remains epigenetically and stably silenced under warm conditions with FT- and SOC1-dependent expression. The FT protein forms a heterodimer with FD, which promotes flowering by activating the meristem identity gene [27,28]. FLC directly restrained SOC1 through a CArG box on the SOC1 promoter that is activated on long days through a separate cis element, but vernalization restrains FLC. However, vernalization activates SOC1 expression during long days. In addition, during short days vernalization activates SOC1 because the positive factor of vernalization may induce flowering [29].

## 5. Conclusions

Vernalization could induce off-season flowering, increasing the speed of flowering via the interaction between low temperature and hour of vernalization. The vernalization could also upregulate expression of the three flowering genes VRN5, FT and SOC1, implying their participation in the flowering of sub-tropical strawberry cv. Pharachatan 80 in relation to increasing the percentage of flowering, the number of inflorescences and the number of flowers produced, all which were higher than in non-vernalized plants. The vernalization treatment can be applied to the off-season strawberry production industry and provides gene expression data for sub-tropical strawberry molecular breeding.

**Author Contributions:** Conceptualization, T.T. and D.N.; methodology, T.T., T.P. and D.N.; software, T.T.; validation, W.B., T.P. and D.N.; formal analysis, T.T., T.P. and D.N.; investigation, T.T., T.P. and D.N.; data curation, T.T., W.B., T.P. and D.N.; writing—original draft preparation, T.T., W.B. and D.N.; writing—review and editing, T.T., W.B., T.P. and D.N.; visualization, T.T.; supervision, T.T., W.B., T.P. and D.N.; project administration, T.T. and D.N.; funding acquisition, T.T. and D.N. All authors have read and agreed to the published version of the manuscript.

**Funding:** National Research Council of Thailand: GSCMU(NRCT)/08/2565.

**Data Availability Statement:** T.T. and D.N. are responsible for data keeping, and data are available upon request.

**Acknowledgments:** This research and innovation activity was funded by the National Research Council of Thailand (NRCT). We wish to thank the Horticulture Division, Department of Plant and Soil Sciences, Faculty of Agriculture at Chiang Mai University for supporting materials, scientific equipment and places to conduct research. Lastly, thanks to the Graduate school of Chiang Mai University for supporting the teaching assistant and research assistant (TA/RA) scholarships.

**Conflicts of Interest:** The authors declare no conflict of interest.

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
