# Peer review of "Effects of Vernalization on Off–Season Flowering and Gene Expression in Sub-Tropical Strawberry cv. Pharachatan 80"

_horticulturae, doi:10.3390/horticulturae9010087_

Round 1

Reviewer 1 Report

Kindly find review report in attachments

Reviewer 2 Report

General comments

The article entitled “Effects of Vernalization on Off-season Flowering and Gene Expression in Tropical Strawberry cv. Phrarachatan 80” refers to the expression of genes related to strawberry flowering and the effect of various vernalization conditions on flowering physiology. The work is of relevant cognitive importance and a practical aspect, important for horticultural production. The research methodology, statistical elaboration of results and conclusions do not raise any objections.

Detailed comments

Before publishing the article, some additions are recommended. They mainly concern the taxonomic affiliation of the Phrarachatan 80 cultivar, referred to as "tropical strawberry". Is it a Fragaria x ananassa cultivar? Information on the origin of the variety used in the research should be included in the methodology. The more, so that the Chilean white strawberry (probably F. chiloensis) is mentioned in the discussion.
